# Acute Effects of Supra- and High-Loaded Front Squats on Mechanical Properties of Lower-Limb Muscles

**DOI:** 10.3390/sports11080148

**Published:** 2023-08-02

**Authors:** Michal Krzysztofik, Michal Wilk, Dominik Kolinger, Anna Pisz, Katarzyna Świtała, Jan Petruzela, Petr Stastny

**Affiliations:** 1Department of Sport Games, Faculty of Physical Education and Sport, Charles University in Prague, 110 00 Prague, Czech Republic; m.wilk@awf.katowice.pl (M.W.);; 2Institute of Sport Sciences, The Jerzy Kukuczka Academy of Physical Education, 40-065 Katowice, Poland; 3Faculty of Physical Education, Gdansk University of Physical Education and Sport, 80-336 Gdańsk, Poland; katarzyna.switala@awf.gda.pl

**Keywords:** resistance training, fatigue, eccentric, myotonometry, muscle tone, stiffness

## Abstract

Knowledge about the acute effects of supramaximal-loaded resistance exercises on muscle mechanical properties is scarce. Therefore, this study aimed to examine changes in dominant limb biceps femoris and vastus lateralis oscillation frequency and stiffness before and after high- and supramaximal-loaded front squats. Nineteen male handball players participated in the experimental session with a barbell front squat 1RM. The first set was performed at 70% of the 1RM for four repetitions, and the second and third sets were performed at 90%1RM in an eccentric–concentric or an eccentric-only manner at 120% of the 1RM for three repetitions. The handheld myometer was used for the measurement of the biceps femoris and vastus lateralis stiffness and the oscillation frequency of the dominant limb 5 min before and at the 5th and 10th min after front squats. A two-way ANOVA neither indicated a statistically significant interaction (*p* = 0.335; η^2^ = 0.059 and *p* = 0.103; η^2^ = 0.118), the main effect of a condition (*p* = 0.124; η^2^ = 0.126 and *p* = 0.197; η^2^ = 0.091), nor the main effect of the time point (*p* = 0.314; η^2^ = 0.06 and *p* = 0.196; η^2^ = 0.089) for vastus lateralis and biceps femoris stiffness. However, there was a statistically significant interaction (F = 3.516; *p* = 0.04; η^2^ = 0.163) for vastus lateralis oscillation frequency. The post hoc analysis showed a significantly higher vastus lateralis oscillation frequency at POST (*p* = 0.037; d = 0.29) and POST_10 (*p* = 0.02; d = 0.29) compared to PRE during the SUPRA condition. Moreover, Friedman’s test indicated statistically significant differences in biceps femoris oscillation frequency (test = 15.482; *p* = 0.008; Kendall’s W = 0.163). Pairwise comparison showed a significantly lower biceps femoris oscillation frequency in POST (*p* = 0.042; d = 0.31) and POST_10 (*p* = 0.015; d = 0.2) during the HIGH condition compared to that in the corresponding time points during the SUPRA condition. The results of this study indicate that the SUPRA front squats, compared to the high-loaded ones, cause a significant increase in biceps femoris and vastus lateralis oscillation frequency.

## 1. Introduction

The effects of eccentric contractions on physiological changes are of great interest to researchers, primarily due to their unique properties such as the ability to apply supramaximal loads (significantly exceeding those possible to apply during concentric and isometric contractions) [1]. Therefore, many studies focus on comparing submaximal and supramaximal loads or different contraction types, such as eccentric-only or coupled eccentric–concentric contractions, performed during exercises [2,3]. It has been shown that these aspects can have an acute impact on the level of induced fatigue, muscle damage, and thus post-exercise performance [3,4]. For example, Ulrich and Parstorfer [5] showed that eccentric-only bench pressing at a 120% one-repetition maximum (1RM) did not enhance subsequent bench press throws. On the contrary, Ong et al. [6], and Krzysztofik et al. [1] showed that exercises performed in an eccentric-only manner (leg press and bench press) and supramaximal external load acutely increase the power output during sports movements (countermovement jump and bench press throw), as long as the load exceeds 100% of the 1RM. Supramaximal loads have the potential to reduce inhibitory reflexes and increase rate coding to a greater extent when compared to submaximal loads [2]. Additionally, the energy requirements for eccentric contractions are approximately four times smaller than those for the same exercise performed concentrically [7]. However, if a higher load is used during a eccentric contraction compared to that used in the concentric condition, fatigue levels could actually be similar between the two. Therefore, it is reasonable to consider that there could be a higher level of acute activation for a given level of fatigue, while also improving the balance between activation and fatigue, ultimately leading to improved performance after exercising [8]. Consequently, activation due to eccentric contractions may induce a relatively higher post-activation performance enhancement (PAPE) effect compared to other types of contractions due to increased activation and either lower or similar levels of fatigue [9].

The monitoring of exercise-induced fatigue is increasingly recognized as an essential aspect in optimizing the dose–response relationship and, ultimately, enhancing performance. One of the simple and potentially useful solutions to this is myotonometry. Myotonometry involves applying a small amount of force to a muscle and measuring its response, i.e., oscillation frequency and stiffness. Specifically, according to the manufacturer, the oscillation frequency characterizes the intrinsic tension of biological soft tissues on the cellular level, while stiffness describes the resistance of such tissue to a force of deformation. Kablan et al. [10] revealed that the stiffness and oscillation frequency obtained is showed a tissue tone with larger values indicating greater muscle tone. That, in turn, might be associated, as demonstrated by Korhonen et al. [11], with increased intramuscular fluid pressure. As discussed by Sejersted and Hargens [12] or by Sleboda and Roberts [13], an increase in intramuscular fluid pressure can substantially affect muscle function by altering the geometry of muscle fibers and, as a result, the levels of generated force. For Instance, Klich et al. [14] reported an acute increase in the stiffness of the vastus lateralis muscle after all-out 200 m and 4000 m track cycling, with a greater increase observed after the shorter distance. Additionally, Trybulski et al. [15] observed a tendency for increased muscle stiffness in the long head of the triceps brachii, accompanied by a decrease in barbell velocity during the bench press exercise. The observed fatigue may be attributed to a higher intramuscular fluid pressure, which could impede the clearance of metabolic byproducts. On the other hand, Krzysztofik et al. [16] demonstrated a decrease in the stiffness of the vastus lateralis muscle concurrent with improvements in countermovement jump height after performing low-volume, high-loaded back squats (three sets of three repetitions at 85% of the 1RM and one to two sets until a 10% velocity loss was reached at 60% of the 1RM). However, the changes in muscle stiffness and oscillation frequency following supramaximal-loaded exercise and that performed in an eccentric-only manner have not been examined. It seems that eccentric contractions may cause both immediate and delayed greater intramuscular pressure compared to those caused by concentric contractions. For instance, a study by Friden et al. [17] demonstrated a significantly higher intramuscular pressure during eccentric exercise than that during concentric exercise, despite the same load being applied. This higher intramuscular pressure was also observed during rest two days later.

To the best of the authors’ knowledge, no study to date has compared the acute effect of supramaximal- and high-loaded front squats on changes in muscle mechanical properties using myotonometry. Examining minor alterations in muscle stiffness and muscle oscillation frequency could be useful in detecting the initial onset of muscle fatigue. This approach has been highlighted in previous studies [10,16,18]. Fast and adequate recovery after exercise plays a crucial role in enhancing performance and reducing the risk of injuries [19]. Therefore, this could hold significant importance in optimizing the individual PAPE effect as it relies on the specific levels of fatigue and potentiation induced. Consequently, it could be utilized to determine the appropriate rest period after exercise, thus minimizing the impact of elevated intramuscular fluid pressure on subsequent performance. Therefore, the purpose of this study was to examine changes in the dominant limb’s biceps femoris and vastus lateralis oscillation frequency and stiffness before and after high- and supramaximal-loaded front squats. Since the acute enhancement in athletic performance was previously observed after similar conditions (i.e., low-volume conditioning at 90% of the 1RM as well as 110 and 130% of the 1RM) [1,16,20,21], it was hypothesized that none of the examined conditions would significantly alter the studied muscles’ mechanical properties.

## 2. Materials and Methods

### 2.1. Experimental Approach to the Problem

This cross-sectional randomized research consisted of one familiarization and two experimental sessions (Figure 1). The cross-sectional approach was used to describe the acute changes in the dominant limb’s biceps femoris and vastus lateralis oscillation frequency and stiffness after front squats, which differed in training intensity.

### 2.2. Participants

Nineteen handball players (age 16.6 ± 1.2 years; body mass 79.7 ± 9.9 kg; height 184 ± 6.3 cm; personal body fat 10.2 ± 5.2%; 3RM 74.3 ± 19.1 kg) of the first Czech national league in the U19 category participated in the experiment. The following criteria were applied for participant selection: (i) absence of neuromuscular and musculoskeletal disorders, (ii) a minimum of two years of experience in resistance training, and (iii) engagement in regular handball and resistance training for at least one year prior to the study. Participants were instructed to maintain their usual dietary and sleep patterns and refrain from consuming any stimulants or alcoholic beverages throughout the study. Additionally, they were advised not to engage in any additional resistance exercises within 48 h before testing to prevent fatigue. Participants had the option to withdraw from the experiment at any time and were provided with information regarding the potential risks and benefits of the study before giving their written informed consent. The study protocol received approval from Bioethics Committee at the Faculty of Physical Education and Sport, Charles University, no EK267/2020, and adhered to the ethical standards outlined in the 2013 Declaration of Helsinki.

### 2.3. Familiarization Session

The session began with a non-specific warm-up consisting of 5 min of stationary cycling at a moderate intensity, followed by bodyweight exercises (10 repetitions each): forward and side bending in standing, side-to-side and back and forward leg swings standing on a single leg, single knee-to-chest stretches, heel-to-glute-with-arm reaches, squats and 5 countermovement jumps with subjective 70% effort. This was followed by 1RM front squat testing. The participants performed 10, 6, 4, and 3 repetitions of the front squat, at a load of 20 kg and 40, 60, and 80% of the self-estimated 1RM, respectively. Next, the load was increased by 2.5–5 kg for each subsequent attempt. An attempt was considered unsuccessful if the participants did not become parallel to the bottom part of the deep squat position. The highest load completed without any help from the spotters was defined as the 1RM. Five-minute rest intervals were allowed between the 1RM attempts, and all 1RM values were obtained within 5 attempts. Moreover, all participants were familiarized with an upcoming experimental session, including a muscle mechanical property assessment.

### 2.4. Experimental Session

After the warm-up (same as during the familiarization session) and baseline muscle mechanical property assessments, the participants performed a conditioning activity in the form of 3 sets of 3–4 repetitions of barbell front squats with loads corresponding to 70–90% of the 1RM. The first set was performed at 70% of the 1RM for 4 repetitions, and the second and third sets were performed at 90% of the 1RM in an eccentric-concentric or an eccentric-only manner at 120% of the 1RM for 3 repetitions. The rest interval between each set was set at 3 min. Then, at the 5th and 10th min after front squats, the muscle mechanical property assessments were repeated. Those time points were selected since previous studies showed that changes in muscle mechanical properties last up to ~10 min following low-volume exercises [10,22].

The experimental sessions were separated by a minimum of 72 h and a maximum of 7 days as the rest period. Participants were instructed to skip heavy lower-body strength training 48 h before each session.

### 2.5. Muscle Mechanical Property Assessment

The handheld myometer (MyotonPRO, Myoton AS, Tallinn, Estonia) was used for the measurement of the biceps femoris’ and vastus lateralis’ stiffness (the resistance of biological soft tissue to a force of deformation in Newtons per meter (N/m)) and oscillation frequency (the intrinsic tension of biological soft tissues on the cellular level in hertz (Hz)) of the dominant limb through superficial mechanical deformation [23]. Previous studies indicated that Myoton is a valid and reliable device with which to quantify muscle tone and stiffness [24,25,26]. The device’s probe was placed perpendicularly to the skin, and brief mechanical compression was applied (0.4 N for 15 milliseconds), with a constant preload force of 0.18 N. After a mechanical impulse, the soft tissue responds in the form of a dampened oscillation, which is measured by an in-built accelerometer, sampled at 3200 Hz [27]. The measurement for the biceps femoris was at 50% of the distance from the isocenter to the lateral epicondyle of the femur [28], while for vastus lateralis it was at 50% of the straight-line distance between the greater trochanter and fibulae capitulum [24]. A permanent marker was used to mark the measurement point. Moreover, the participants were instructed to correct the mark if it started to blur. Measurements were made in a state of muscle relaxation, with participants lying in a prone (biceps femoris) or supine (vastus lateralis) position, by the same researcher at every time point.

### 2.6. Statistical Analysis

All data were analyzed using IBM SPSS Statistics for Macintosh, Version 25.0 (IBM Corp., Armonk, NY, USA) and were shown as means with standard deviations (±SD) with their 95% confidence intervals (CI). Statistical significance was set at *p* < 0.05. The normality of data distribution was verified using Shapiro–Wilk tests and Mauchly’s test was used to test the assumption of sphericity. The relative (two-way mixed effects, absolute agreement, and single-rater intraclass correlation coefficient) and absolute (coefficient of variation) reliability were calculated. The thresholds for interpreting intraclass correlation coefficient results were <0.5 as “poor”, 0.5–0.75 as “moderate”, <0.76–0.9 as “good”, and >0.90 as “excellent” [29], while the coefficient of variation results was <10% as “very good”, 10–20% as “good”, <21–30% as “acceptable”, and >30% as “not acceptable” [30]. The two-way ANOVAs (or nonparametric equivalent test) (2 [HIGH; SUPRA] × 3 time points [baseline; post; post_10]) were used to investigate the influence of supramaximal- and high-loaded front squats on muscle mechanical property variables. The effect size for ANOVA was determined via partial eta squared (η^2^), with values classified as small (0.01–0.059), moderate (0.06–0.137), and large (>0.137) [31]. When a significant interaction or main effect was found, post hoc tests with Bonferroni correction were used to analyze the pairwise comparisons. The magnitude of mean differences was expressed with a standardized effect size (ES). Thresholds for qualitative descriptors of Cohen’s d were defined as ≤0.20 as a small effect, 0.21 to 0.50 as a moderate effect, 0.51 to 0.80 as a large effect, and >0.80 as a very large effect [31].

## 3. Results

Post hoc power analysis using G*Power version 3.1.9.2 (Dusseldorf, Germany) for the parameters, such as “ANOVA, repeated measures, within factors,” was used as a statistical test (with one group of subjects, two experimental conditions, and three measurements) and the significance level of 0.05 indicated that an effect size of at least 0.31 was needed to achieve a power above 80%.

The Shapiro–Wilk test did show a statistically significant violation of the data distribution for vastus lateralis frequency.

The ICC and CV results calculated on the basis of the baseline values are presented in Table 1.

### 3.1. Vastus Lateralis

Two-way ANOVA indicated neither a statistically significant interaction (F = 1.127; *p* = 0.335; η^2^ = 0.059), the main effect of the condition (F = 2.604; *p* = 0.124; η^2^ = 0.126), nor the main effect of the time point (F = 1.141; *p* = 0.314; η^2^ = 0.06) for stiffness (Table 2).

Friedman’s test indicated statistically significant differences in oscillation frequency (test = 15.482; *p* = 0.008; Kendall’s W = 0.163). Pairwise comparison showed a significantly lower oscillation frequency in the POST (*p* = 0.042; d = 0.31) and POST_10 (*p* = 0.015; d = 0.2) time points during the HIGH condition compared to the corresponding time points during the SUPRA condition.

### 3.2. Biceps Femoris

Two-way ANOVA indicated neither a statistically significant interaction (F = 2.419; *p* = 0.103; η^2^ = 0.118), the main effect of a condition (F = 1.792; *p* = 0.197; η^2^ = 0.091), nor the main effect of the time point (F = 1.762; *p* = 0.196; η^2^ = 0.089) for stiffness (Table 3).

Two-way ANOVA indicated a statistically significant interaction (F = 3.516; *p* = 0.04; η^2^ = 0.163) for oscillation frequency. The post hoc analysis showed a significantly higher oscillation frequency at the POST (*p* = 0.037; d = 0.29) and POST_10 (*p* = 0.02; d = 0.29) time points compared to the BA time point during the SUPRA condition (Table 3).

## 4. Discussion

Knowledge about the acute effects of supramaximal-loaded resistance exercises on muscle mechanical properties is scarce. Therefore, this study aimed to compare the acute effect of high- and supramaximal-loaded front squats on the biceps femoris and vastus lateralis oscillation frequency and stiffness. In contrast to the stated hypothesis, this study revealed a significant increase in the biceps femoris oscillation frequency in the 5th and 10th min after SUPRA front squats compared to baseline values. Moreover, the oscillation frequency of vastus lateralis was significantly lower in the 5th and 10th min after HIGH front squats compared to the corresponding time points after SUPRA front squats. Significant changes were found in the stiffness of both studied muscles. These results show that even low-volume supramaximal-loaded front squats lead to significant changes in the oscillation frequency of the lower limb which may indicate the early onset of fatigue.

Our study confirms Hill et al.’s [32] suggestion that changes in the mechanical properties of muscles are related to the intensity of exercise. The increase in muscle oscillation frequency observed after SUPRA front squats may be related to higher intracellular fluid levels and the associated increase in intramuscular pressure. Such an increase in intracellular fluid levels is one of the physiological factors arising from the accumulation of fluid increasing hydrostatic and osmotic gradients, and ultimately increasing muscle volume and intramuscular pressure, which are related to changes in the mechanical properties of muscles. Elevated intramuscular pressure may impact blood flow, tissue oxygenation, and muscle metabolism, thereby influencing muscle function. When intramuscular pressure increases, muscle recovery is delayed due to impaired blood flow, causing pain and an early onset of muscle fatigue decreasing overall performance [10,15]. These effects could be particularly relevant during activities involving sustained or repetitive muscle contractions, potentially compromising athletic performance and recovery. Therefore, the results of this study could in some way explain the lack of improvement in performance after supramaximal-loaded exercises [5], which could be a consequence of increased intramuscular pressure after their completion. On the other hand, most studies indicate an improvement in fitness after conditioning with similar parameters (low-volume front squats at 110–130% of the 1RM) [1,4,6,20]. The balance between concomitant fatigue and potentiation following exercise determines whether an improvement or decrease in fitness is noted [33]. Therefore, it seems that supramaximal loads cause fatigue, but the potentiation produced could exceed them, which is why some studies have reported performance improvement. At this point, it should be emphasized that the impact of following supramaximal-loaded exercises may depend on participants’ strength levels and training experience. It has been shown that stronger participants with more training experience are more resistant to fatigue, and thus able to achieve a meaningful PAPE effect [33]. For example, in a study by Ulrich and Parstorfer [5] in which a supramaximal load failed to enhance performance, participants’ strength level in bench presses was 1.18 kg·body mass^−1^ while in a study by Krzysztofik et al. [1] which successfully improved athletic performance, it was 1.55 kg·body mass^−1^, whereas in this study the participants’ strength level was 1.00 ± 0.22 kg/BW kg·body mass^−1^ which is considered low [33]. It seems unlikely that the level of strength or training experience can affect the oscillation frequency; however, to the best of our knowledge, there is currently no evidence available with which to compare our study. Therefore, in order to determine this, further studies are needed to assess the impact of supramaximal loads on muscle mechanical properties, performance, and contractile properties in groups of participants with different training experiences. To fully understand the complex interactions between intramuscular pressure and its multifaceted effects on the musculoskeletal system, further research also evaluating changes in blood flow and tissue oxygenation is essential.

Changes in oscillation frequency were similar in the 5th and 10th min after completing the SUPRA front squats. Thus, the significant increase in oscillation frequency induced was not transient; this study indicates that it was sustained for at least 10 consecutive minutes, and during that time may have affected muscle function. To our knowledge, studies that assess the duration of elevated intramuscular pressure are limited to a study by the authors of [17] that showed an increase in intramuscular pressure 2 days after exercise but there is a lack of short-term values, i.e., those from within minutes after exericise. However, it is well known that induced fatigue and muscle damage from supramaximal loads can persist for several days [34,35].

The current study contains several limitations which should be addressed: (1) the study did not include performance assessments, and hence it is not certain that the increase in oscillation frequency was related to the decrease in muscle function (2); the study was conducted only on young, trained men so therefore, the findings may not be generalizable to women and inactive people. Finally, we analyzed only a single supramaximal load (120% of the 1RM) and the volume (two sets of three repetitions) of front squat effects; therefore, the generalizability of these results to the lower supramaximal loads remains unclear.

## 5. Conclusions

The results of this study indicate that SUPRA front squats in comparison to high-loaded ones cause a significant increase in the biceps femoris and vastus lateralis oscillation frequency. These results suggest that even low-volume supramaximal-loaded front squats lead to a significant increase in the oscillation frequency of lower limbs which may indicate the early onset of fatigue. However, further studies are needed to determine whether or not the observed oscillation frequency increase will have a harmful impact on muscle function, i.e., force production.

## Figures and Tables

**Figure 1 sports-11-00148-f001:**
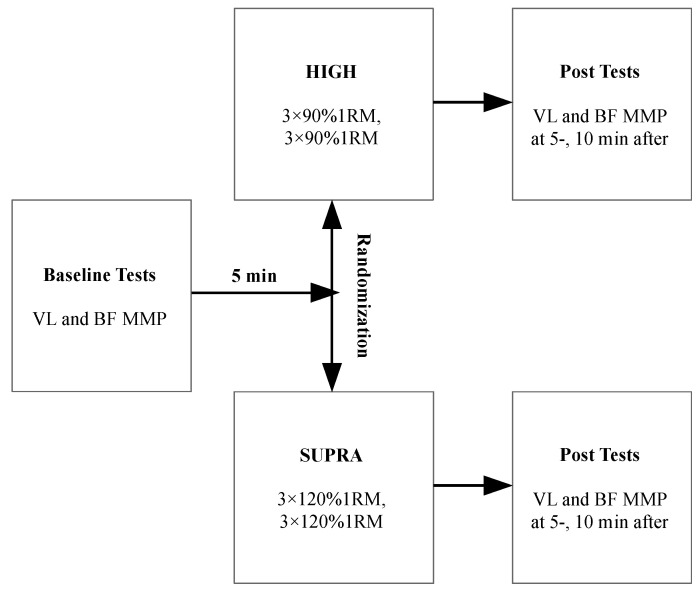
Study design. VL—vastus lateralis; BF—biceps femoris; MMP—muscle mechanical properties; HIGH—high-loaded condition; SUPRA—supramaximal condition; 1RM—one repetition maximum.

**Table 1 sports-11-00148-t001:** Intersession reliability of the analyzed variables.

Variable	ICC (95%CI)	CV (SD)
VL Stiffness	0.94 (0.84–0.98)	3.9 ± 3.1%
VL Frequency	0.89 (0.71–0.96)	3.7 ± 3.2%
BF Stiffness	0.91 (0.77–0.97)	2.6 ± 1.5%
BF Frequency	0.91 (0.77–0.96)	2.1 ± 1.5%

ICC—intraclass correlation coefficient; CV—coefficient of variation; VL—vastus lateralis; BF—biceps femoris; SD—standard deviation.

**Table 2 sports-11-00148-t002:** Comparison of vastus lateralis stiffness and oscillation frequency pre- and post-conditioning activity.

Condition	BA (95%CI)	Post (95%CI)	Post_10 (95%CI)	ES (Pre vs. Post)	ES (Pre vs. Post_10	ES (Post vs. Post_10
Stiffness (N/m)						
SUPRA	316 ± 45 (294 to 337)	325 ± 45 (303 to 346)	321 ± 52 (296 to 346)	0.2	0.1	0.08
HIGH	312 ± 45 (290 to 334)	314 ± 44 (292 to 335)	311 ± 48 (288 to 334)	0.04	0.02	0.06
ES	0.09	0.24	0.2			
**Oscillation Frequency (Hz)**						
SUPRA	17.1 ± 2 (16.2 to 18)	17.7 ± 2 * (16.8 to 18.7)	17.4 ± 2 * (16.5 to 18.4)	0.29	0.15	0.15
HIGH	17.1 ± 1.9 (16.2 to 18.1)	17.1 ± 1.8 (16.2 to 18)	17 ± 1.9 (16.1 to 18)	0.00	0.05	0.05
ES	0.00	0.31	0.2			

BA—baseline; CI—confidence interval, ES—effect size, SUPRA—supramaximal condition, HIGH—high-load condition, *—a significant difference between conditions.

**Table 3 sports-11-00148-t003:** Comparison of biceps femoris’ stiffness and oscillation frequency pre- and post-conditioning activity.

Condition	BA (95%CI)	Post (95%CI)	Post_10 (95%CI)	ES (Pre vs. Post)	ES (Pre vs. Post_10	ES (Post vs. Post_10
Stiffness (N/m)						
SUPRA	299 ± 20 (289 to 308)	307 ± 22 (296 to 318)	305 ± 20 (295 to 314)	0.37	0.29	0.09
HIGH	301 ± 23 (289 to 312)	300 ± 19 (291 to 309)	299 ± 25 (287 to 311)	0.05	0.08	0.04
ES	0.09	0.33	0.26			
**Oscillation Frequency (Hz)**						
SUPRA	16.3 ± 1 (15.8 to 16.7)	16.6 ± 1 * (16.1 to 17.1)	16.6 ± 1 * (16.1 to 17.1)	0.29	0.29	0.00
HIGH	16.4 ± 1 (15.9 to 16.9)	16.4 ± 0.9 (15.9 to 16.8)	16.4 ± 0.9 (15.9 to 16.8)	0.00	0.00	0.00
ES	0.1	0.21	0.21			

BA—baseline; CI—confidence interval, ES—effect size, SUPRA—supramaximal condition, HIGH—high-load condition, *—a significant difference in comparison to baseline in a particular condition.

## Data Availability

The data used to support the findings of this study are available from the corresponding author upon request.

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
