# Peer review of "Acute Effects of Supra- and High-Loaded Front Squats on Mechanical Properties of Lower-Limb Muscles"

_sports, 2023, doi:10.3390/sports11080148_

Round 1

Reviewer 1 Report

Overall, well-designed study with a well-organized and written manuscript. Some additional details are needed. See comments directly on the .pdf.

Author Response

Reviewer #1

Overall, well-designed study with a well-organized and written manuscript. Some additional details are needed. See comments directly on the .pdf.

Reply: We really thank the reviewer for the positive and insightful comments that helped us to improve the quality of the manuscript. We hope that the revised manuscript may satisfactorily address the concerns previously listed. All changes in the text are highlighted in red.

Comment 1: Please define this acronym before using it. (Line 61; PAPE)

Reply: Thank you for pointing this out. Corrected.

Comment 2: Stretching of which muscles? (Line 140)

Reply: Thank you for this comment. We decided to revise this sentence by providing more details.

“The session began with a non-specific warm-up consisting of 5 minutes of stationary cycling at moderate intensity, followed by bodyweight exercises (10 repetitions each): forward and side bending in standing, side-to-side and back and forward leg swings standing on a single leg, single knee-to-chest stretch, heel-to-glute-with-arm-reach, squats and 5 countermovement jumps with subjective 70% effort.”

Comment 3: what is CA? (Line 152)

Reply: We apologize for the confusion. We have omitted this abbreviation.

Comment 4: Please describe (and reference) the results of validation and precision studies on this device. This is especially important because you are doing repeated testing with the device. (Line 162)

Reply:  Thank you for pointing this out. We provided references supporting the validity and reliability of the Myoton device in quantifying muscle tone and stiffness. Moreover, we would like to mention and we also report the intraclass correlation coefficient and coefficient of variation values obtained within our study.

  1. Chen, G.; Wu, J.; Chen, G.; Lu, Y.; Ren, W.; Xu, W.; Xu, X.; Wu, Z.; Guan, Y.; Zheng, Y.; et al. Reliability of a Portable Device for Quantifying Tone and Stiffness of Quadriceps Femoris and Patellar Tendon at Different Knee Flexion Angles. PLOS ONE 2019, 14, e0220521, doi:10.1371/journal.pone.0220521.
  2. Bizzini, M.; Mannion, A.F. Reliability of a New, Hand-Held Device for Assessing Skeletal Muscle Stiffness. Clin. Biomech. 2003, 18, 459–461, doi:10.1016/S0268-0033(03)00042-1.
  3. Pruyn, E.C.; Watsford, M.L.; Murphy, A.J. Validity and Reliability of Three Methods of Stiffness Assessment. J. Sport Health Sci. 2016, 5, 476–483, doi:10.1016/j.jshs.2015.12.001.

Comment 5: what units are used to describe the stiffness measured by this instrument (please reference) (Line 163)

Reply: Thank you very much for this comment. We have revised this sentence and provided a reference:The handheld myometer (MyotonPRO, Myoton AS, Tallinn, Estonia) was used for the measurement of the biceps femoris and vastus lateralis stiffness (the resistance of biological soft tissue to a force of deformation; in Newton per meter [N/m]) and oscillation frequency (the intrinsic tension of biological soft tissues on the cellular level; in hertz [Hz]) of the dominant limb through superficial mechanical deformation.”

Gapeyeva, H.; Vain, A. Methodical Guide: Principles of Applying Myoton in Physical Medicine and Rehabilitation.; Tartu, Estonia: Muomeetria, 2008.

Comment 6: how is the acceleration of a muscle recorded externally? what is the validation and precision of this accelerometer? who is the manufacturer of this accelerometer? (Line 166)

Reply: Thank you for this comment. This is an in-built accelerometer (in Myoton device) with a sampling frequency of 3200 Hz. We revised this sentence to the following one: “After a mechanical impulse, the soft tissue responds in the form of a dampened oscillation, which is measured by an in-built accelerometer, sampled at 3200 Hz.”

Ditroilo, M.; Hunter, A.M.; Haslam, S.; De Vito, G. The Effectiveness of Two Novel Techniques in Establishing the Mechanical and Contractile Responses of Biceps Femoris. Physiol. Meas. 2011, 32, 1315–1326, doi:10.1088/0967-3334/32/8/020.

Comment 7: please reference this statement (Line 191)

Reply: Thank you for pointing this out. Reference added: Cohen, J. Statistical Power Analysis for the Behavioral Sciences.; Elsevier Science: Burlington, 2013; ISBN 978-1-4832-7648-9.

Comment 8: is this newton meters? please define this somewhere. (Line 217)

Reply: Thank you for this comment. In the Muscle Mechanical Properties Assessment paragraph, we provided a definition: “…the resistance of biological soft tissue to a force of deformation; in Newton per meter [N/m]...”

Comment 9: In addition to restating your purpose, please briefly remind the reader what the need/rationale for this study was? (Line 234)

Reply: Thank you for this comment. We have added the following sentence: “Knowledge about the acute effects of supramaximal-loaded resistance exercises on muscle mechanical properties is scarce.”

Reviewer 2 Report

This study aimed to investigate changes in oscillation frequency and stiffness of the dominant limb's biceps femoris and vastus lateralis before and after high- and supramaximal-loaded front squats. Nineteen male handball players participated in the experiment. Results showed no statistically significant interaction or main effect of condition or stiffness. However, there was a significant interaction for vastus lateralis oscillation frequency, which increased significantly during the SUPRA condition compared to the HIGH condition. The study suggests that SUPRA front squats lead to higher oscillation frequency in biceps femoris and vastus lateralis.

The present study focuses on an interesting topic and shows potential. However, after careful review, several critical points have come to light that the authors should consider.

Firstly, the experimental hypothesis states that evaluating muscle mechanical changes using Myotone (muscle stiffness and oscillation frequency) could provide valuable information about muscle fatigue and potential performance enhancement (PAPE). However, the initial hypothesis was stated as ".... it was hypothesized that none of the examined conditioning would significantly alter studied muscles' mechanical properties." This appears somewhat contradictory. Therefore, it is unclear what information the study aims to provide. Additionally, the authors mention the lack of analysis regarding the possible presence of PAPE as a limitation, which significantly impacts the discussion.

Secondly, the variation in oscillation frequency merits further consideration. While statistical significance may support the observed variations, it is essential to assess whether such changes are practically meaningful (within the signal's variability). From a physiological perspective, would an increase in intramuscular pressure lead to more pronounced changes? The authors should elaborate on this aspect to provide better clarity.

There are also some minor points to consider:

1.     Why is it useful to specify in the inclusion criteria that participants should have been handball players for at least one year?

2.     Line 152: What does "CA" stand for?

3.     Lines 179-181: What measurements were used to calculate repeatability?

4.     How was the partial eta square interpreted?

5.     Please indicate in Table 2 the significant difference between POST supra and high in the vastus lateralis.

6.     In Table 3, what do "Slow and Fast" and "Supra and High" refer to?

7.     Lines 259-261: Therefore, it seems that supramaximal loads cause fatigue, but the potentiation produced COULD exceed them, which is why some studies have reported performance improvement.

Author Response

Reviewer #2

This study aimed to investigate changes in oscillation frequency and stiffness of the dominant limb's biceps femoris and vastus lateralis before and after high- and supramaximal-loaded front squats. Nineteen male handball players participated in the experiment. Results showed no statistically significant interaction or main effect of condition or stiffness. However, there was a significant interaction for vastus lateralis oscillation frequency, which increased significantly during the SUPRA condition compared to the HIGH condition. The study suggests that SUPRA front squats lead to higher oscillation frequency in biceps femoris and vastus lateralis.

The present study focuses on an interesting topic and shows potential. However, after careful review, several critical points have come to light that the authors should consider.

Reply: We really thank the reviewer for the positive and insightful comments that helped us to improve the quality of the manuscript. We hope that the revised manuscript may satisfactorily address the concerns previously listed. All changes in the text are highlighted in red.

Firstly, the experimental hypothesis states that evaluating muscle mechanical changes using Myotone (muscle stiffness and oscillation frequency) could provide valuable information about muscle fatigue and potential performance enhancement (PAPE). However, the initial hypothesis was stated as ".... it was hypothesized that none of the examined conditioning would significantly alter studied muscles' mechanical properties." This appears somewhat contradictory. Therefore, it is unclear what information the study aims to provide. Additionally, the authors mention the lack of analysis regarding the possible presence of PAPE as a limitation, which significantly impacts the discussion.

Reply: Thank you for this comment. We are aware that this is a significant limitation of our study, so we tried to be restrained in drawing conclusions.

Secondly, the variation in oscillation frequency merits further consideration. While statistical significance may support the observed variations, it is essential to assess whether such changes are practically meaningful (within the signal's variability). From a physiological perspective, would an increase in intramuscular pressure lead to more pronounced changes? The authors should elaborate on this aspect to provide better clarity.

Reply: Thank you for this comment. We have added the following sentences: Elevated intramuscular pressure may impact blood flow, tissue oxygenation, and muscle metabolism, thereby influencing muscle function. When intramuscular pressure increases, muscle recovery is delayed due to impaired blood flow, causing pain and early onset of muscle fatigue decreasing overall performance [10,15]. These effects could be particularly relevant during activities involving sustained or repetitive muscle contractions, potentially compromising athletic performance and recovery.” andTo fully understand the complex interactions between intramuscular pressure and its multifaceted effects on the musculoskeletal system further research evaluating also changes in blood flow and tissue oxygenation is essential.”

There are also some minor points to consider:

  1. Why is it useful to specify in the inclusion criteria that participants should have been handball players for at least one year?

Reply: Thank you for this comment. We aimed to highlight that participants should be regularly involved training process, in this scenario, in handball and resistance training.

  1. Line 152: What does "CA" stand for?

Reply: Thank you for pointing this out. We removed this abbreviation and provide full words for CA which stated “conditioning activity”.

  1. Lines 179-181: What measurements were used to calculate repeatability?

Reply: Thank you for this comment. We added that ICC and CV were calculated from baseline values: “The ICC and CV results calculated on the basis of the baseline values are presented in Table 1.”

  1. How was the partial eta square interpreted?

Reply: Thank you for this comment. We have added the following sentence to the Statistical Analysis section: The effect size for ANOVA was determined by partial eta squared (η2), with values classified as small (0.01–0.059), moderate (0.06–0.137), and large (>0.137) [29].”

  1. Please indicate in Table 2 the significant difference between POST supra and high in the vastus lateralis.

Reply: Thank you for this comment. We highlighted the statistically significant difference with an asterisk.

  1. In Table 3, what do "Slow and Fast" and "Supra and High" refer to?

Reply: Thank you for pointing this out. We apologize for this typo, a correction has been made.

  1. Lines 259-261: Therefore, it seems that supramaximal loads cause fatigue, but the potentiation produced COULD exceed them, which is why some studies have reported performance improvement.

Reply: Thank you very much for this comment. A correction has been made.

Reviewer 3 Report

In general, you should state the gap in the literature and what your study will add, clarify the method of muscle contraction you used in the methodology, indicate the reliability of the instruments used. It's a major downside that you didn't measure performance. How do we know if athletes' performance will be affected?

Comments can be found in the attachment

Author Response

Reviewer #3

In general, you should state the gap in the literature and what your study will add, clarify the method of muscle contraction you used in the methodology, indicate the reliability of the instruments used. It's a major downside that you didn't measure performance. How do we know if athletes' performance will be affected?

Reply: We really thank the reviewer for the positive and insightful comments that helped us to improve the quality of the manuscript. We hope that the revised manuscript may satisfactorily address the concerns previously listed. All changes in the text are highlighted in red.

Comment 1: Indicate what kind of muscle contraction (e.g. eccentric) (Line 21)

Reply: Thank you for pointing this out. We have specified that in this case the front squat was performed in an eccentric-only manner.

Comment 2: The first time analyzed the abbreviation (PAPE; Line 61)

Reply: Thank you for pointing this out. Corrected.

Comment 3: What is the use of your study? What gap will it fill and what will it add to the scientific community and the coaches? (Line 103)

Reply: Thank you for this comment. To strengthen the knowledge value resulting from our study on the scientific community we revised the last paragraph of the Introduction to the following one: “Examining minor alterations in muscle stiffness and muscle oscillation frequency could be useful in detecting the initial onset of muscle fatigue. This approach has been highlighted in previous studies [10,16,18]. Fast and adequate recovery after exercise plays a crucial role in enhancing performance and reducing the risk of injuries [19]. Therefore, this could hold significant importance in optimizing the individual PAPE effect as it relies on the specific levels of fatigue and potentiation induced. Consequently, it could be utilized to determine the appropriate rest period after exercise, thus minimizing the impact of elevated intramuscular fluid pressure on subsequent performance.”

Comment 4: explain why the measurements were taken at 5 and 10 minutes after (you relied on a previous study); (Figure 1)

Reply: Thank you for this comment. We explained the selection of such time points: Then, at the 5th and 10th min after front squats, the muscle mechanical properties assessments have been repeated. Those time points have been selected since previous studies showed that changes in muscle mechanical properties last up to ~10 min following low-volume exercises [10,21].

Kablan, N.; Alaca, N.; Tatar, Y. Comparison of the Immediate Effect of Petrissage Massage and Manual Lymph Drainage Following Exercise on Biomechanical and Viscoelastic Properties of the Rectus Femoris Muscle in Women. J. Sport Rehabil. 2021, 30, 725–730, doi:10.1123/jsr.2020-0276.

Krzysztofik, M.; Spieszny, M.; Trybulski, R.; Wilk, M.; Pisz, A.; Kolinger, D.; Filip-Stachnik, A.; Stastny, P. Acute Effects of Isometric Conditioning Activity on the Viscoelastic Properties of Muscles and Sprint and Jumping Performance in Handball Players. J. Strength Cond. Res. 2023, 37, 1486–1494, doi:10.1519/JSC.0000000000004404.

Comment 5: The first time analyzed the abbreviation (Line 152)

Reply: We apologize for the confusion. We have omitted this abbreviation.

Comment 6: indicate the type of muscle contraction. You can't talk about 3 reps in concentric contraction with 120% of RM unless there's underway (which I didn't understand). The same should be mentioned in the abstract. (Line 156)

Reply: Thank you for pointing this out. We have specified that in this case the front squat was performed in an eccentric-only manner.

Comment 7: indicate the reliability and validity of the measurement and instrument (Line 161)

Reply: Thank you for pointing this out. We provided references supporting the validity and reliability of the Myoton device in quantifying muscle tone and stiffness. Moreover, we would like to mention and we also report the intraclass correlation coefficient and coefficient of variation values obtained within our study.

  1. Chen, G.; Wu, J.; Chen, G.; Lu, Y.; Ren, W.; Xu, W.; Xu, X.; Wu, Z.; Guan, Y.; Zheng, Y.; et al. Reliability of a Portable Device for Quantifying Tone and Stiffness of Quadriceps Femoris and Patellar Tendon at Different Knee Flexion Angles. PLOS ONE 2019, 14, e0220521, doi:10.1371/journal.pone.0220521.
  2. Bizzini, M.; Mannion, A.F. Reliability of a New, Hand-Held Device for Assessing Skeletal Muscle Stiffness. Clin. Biomech. 2003, 18, 459–461, doi:10.1016/S0268-0033(03)00042-1.
  3. Pruyn, E.C.; Watsford, M.L.; Murphy, A.J. Validity and Reliability of Three Methods of Stiffness Assessment. J. Sport Health Sci. 2016, 5, 476–483, doi:10.1016/j.jshs.2015.12.001.

Comment 8: add the names of the researchers or delete the 'by'' (Line 266 and 267)

Reply: Thank you for this comment. A correction has been made.

Round 2

Reviewer 2 Report

The Authors replied adequately to all my previous suggestions.

Reviewer 3 Report

ok